

# Toward density functional theory on quantum computers?

**Bruno Senjean[1]⋆, Saad Yalouz[2]† and Matthieu Saubanère[1]‡**

**1** ICGM, Université de Montpellier, CNRS, ENSCM, Montpellier, France
**2** Laboratoire de Chimie Quantique, Institut de Chimie, CNRS/Université de Strasbourg,
4 rue Blaise Pascal, 67000 Strasbourg, France

⋆ bruno.senjean@umontpellier.fr , † yalouzsaad@gmail.com ,
‡ matthieu.saubanere@umontpellier.fr

## Abstract

Quantum Chemistry and Physics have been pinpointed as killer applications for quantum computers, and quantum algorithms have been designed to solve the Schrödinger equation with the wavefunction formalism. It is yet limited to small systems, as their size is dictated by the number of qubits available. Computations on large systems rely mainly on mean-field-type approaches such as density functional theory, for which no quantum advantage has been envisioned so far. In this work, we question this *a priori* by proposing a counter-intuitive mapping from the non-interacting to an auxiliary interacting Hamiltonian that may provide the desired advantage.



# 1  Introduction

Quantum computers have shown promises to solve specific problems that are currently intractable on classical computers, but they are still in their infancy to solve problems of industrial interest faster than classical computers [1, 2]. One of the nearest-term application of quantum computers is quantum chemistry (see Refs. [3–7] and references therein), where the focus is on wavefunction theory (WFT) that targets a numerically exact solution of the electronic structure problem. While quantum phase estimation (QPE) algorithms are in principle capable of solving the problem in its entirety [8–12], the required circuit-depth precludes their application in the noisy intermediate-scale quantum (NISQ) era [13]. More efficient algorithms have therefore been developed such as the quantum stochastic drift protocol [14], or the direct simulation of the Hamiltonian using linear combination of unitaries and the qubitization formalism [15–18]. More adapted to the NISQ era, several variational quantum algorithms (hybrid quantum-classical) have been specifically designed to prepare ground states [19–23] and, more recently, excited states [24–26], and to calculate atomic forces and molecular properties [27–30].

However, despite the exponential speed-up announced by quantum computers, it remains unclear when practical quantum advantage will really be met in practice, and expecting any high impact applications in a near future is difficult [31–34]. Indeed, the applications of quantum algorithms for quantum chemistry remain limited in terms of size of affordable systems, as the size of the system dictates the number of required qubits. Even though the number of qubits on quantum devices is expected to increase rapidly, stable machines able to tackle real quantum chemistry systems are not expected in the next few years. Within noisy quantum computers of the NISQ era, a high precision outcome is illusive and the quest for chemical precision remains a long journey for relevant applications of high societal and industrial impact.

Currently, classical computations on large systems from chemistry, condensed matter physic, and even biology, rely mainly on density functional theory (DFT) [35, 36], for which no quantum advantage is *a priori* expected as it only scales cubically with respect to the system size. Instead, recent works focus on the challenging construction of accurate exchange-correlation (XC) density functionals – for which the precise determination is QMA-hard [37] – using matrix product states, machine learning and quantum computers [38–43]. Solving the Kohn–Sham potential inversion problem where the density of the time-evolved many-body system is measured on a quantum computer has also been investigated [44–46]. Other interesting works have generalized the Hohenberg–Kohn and Runge–Gross theorems of DFT and its time-dependent version, respectively, to qubit Hamiltonians, thus opening the possibility to approximate many-body observables in quantum computing as single-qubit quantities functionals of the density [47, 48]. But none of the aforementioned works aim to solve the Kohn–Sham (KS) non-interacting problem on a quantum computer. Only few attempts have been done to perform mean-field approximations on quantum computers, such as Hartree–Fock with the experimental milestone on a 12 qubits platform [49], or the calculation of the one-particle density matrix on quantum annealers [50]. In both cases, no pratical quantum advantage has been envisioned. Hence, DFT remains applied on classical computers, although sometimes combined with WFT on quantum computers by using embedding strategies [6, 51, 52].

In this work, we investigate the benefit of using digital quantum computers to scale up mean-field-type methods such as DFT. A possible quantum advantage is discussed in terms of a counter-intuitive mapping between the KS Hamiltonian and an auxiliary interacting one, expressed in the computational basis, as opposed to what has been done for decades. With such new encoding, mean-field type Hamiltonians can, in some ideal cases, be solved with an exponential speed-up on quantum computers, in analogy with interacting Hamiltonians.

Besides, DFT has its own level of approximations that leads to errors of orders of magnitudes higher than chemical accuracy. As such, one can accept errors that are of the order of the DFT approximations instead of chemical accuracy, and less efforts in applying quantum error correction codes and error mitigation techniques should be required. Hence, the noise of quantum computers is expected to play a less significant impact on the outcome of the calculation. This is a key observation that could bring DFT to center stage as the nearest-term application of quantum computers for molecular, material and biological science.

## 2 A quantum algorithm for Density Functional Theory

The undeniable success of DFT relies on the transformation of the complex many-body quantum electronic problem into an effective and electronic density-dependent one-body problem [35,36]. More precisely, the ground-state energy and ground-state electronic density of a given complex system can be obtained exactly by solving the KS equation [36]:

$$\left(\hat{T} + \hat{v}^{\text{KS}}[\mathbf{n}]\right)\left|\Phi^{\text{KS}}\right\rangle = \mathcal{E}^{\text{KS}}\left|\Phi^{\text{KS}}\right\rangle, \tag{1}$$

where $\hat{T}$ is the kinetic energy operator, $\left|\Phi^{\text{KS}}\right\rangle$ is a single Slater determinant and

$$\hat{v}^{\text{KS}}[\mathbf{n}] = \hat{v}^{\text{ext}} + \hat{v}^{\text{H}}[\mathbf{n}] + \hat{v}^{\text{xc}}[\mathbf{n}], \tag{2}$$

is the density-dependent KS potential operator. It contains the external potential $\mathbf{v}^{\text{ext}}(\vec{r})$, i.e. the ion-electron interaction, the trivial Hartree potential $\mathbf{v}^{H}[\mathbf{n}](\vec{r})$ and finally the so-called exchange and correlation (XC) potential $\mathbf{v}^{\text{xc}}[\mathbf{n}](\vec{r})$. The latter contains all non-trivial contributions arising from the electron-electron interaction.

The KS equation (1) is complemented by the self-consistent condition to obtain the electronic density

$$\mathbf{n}(\vec{r}) = \sum_{k=1}^{N_{\text{occ}}} \langle \vec{r}|\varphi_k\rangle\langle\varphi_k|\vec{r}\rangle, \tag{3}$$

where $\{\varphi_k(\vec{r})\}$ are the KS orbitals in $\left|\Phi^{\text{KS}}\right\rangle$, $N_{\text{occ}}$ denotes the number of occupied spin-orbitals and $\vec{r}$ the position vector. The electronic density is then used to compute a new KS potential, so that Eqs. (1) and (3) are self-consistently and iteratively solved until convergence is reached. Providing the exact XC functional leads to equivalent electronic density between the non-interacting auxiliary KS system and the ground state of the physical system $\mathbf{n}_0$, i.e. $\mathbf{n}^{\Phi^{\text{KS}}}(\vec{r}) = \mathbf{n}_0(\vec{r})$. The exact ground-state energy of the physical system is also recovered as follows [36]:

$$E_0 = \mathcal{E}^{\text{KS}} + E_{\text{Hxc}}[\mathbf{n}^{\Phi^{\text{KS}}}] - \int \mathbf{v}^{\text{Hxc}}[\mathbf{n}^{\Phi^{\text{KS}}}](\vec{r})\mathbf{n}^{\Phi^{\text{KS}}}(\vec{r})\,d\vec{r}, \tag{4}$$

where $\mathcal{E}^{\text{KS}} = \sum_{k=1}^{N_{\text{occ}}} \varepsilon_k$ and $\varepsilon_k$ denotes the $k$-th KS orbital energy. Despite plenty of different flavors of approximations, the exact XC functional is in general not known thus leading to subsequent density- and functional-driven errors [53].

### 2.1 From Kohn–Sham to auxiliary interacting Hamiltonian mapping

The key challenge under any quantum algorithm is to encode the problem in a comprehensive language for the quantum computer. In the context of WFT, the encoding of an interacting

system described in a basis set of $N$ spin-orbitals generally requires a quantum computer with $N$ qubits (for instance using the Jordan–Wigner (JW) [54] or Bravyi–Kitaev [55] transformations). It implies a one-to-one correspondence between the state of the qubits and the occupation of the spin-orbitals. More precisely, a qubit in state $|0\rangle$ (respectively $|1\rangle$) is interpreted as an empty (respectively occupied) spin-orbital. Hence, each of the $2^N$ bitstrings generated by the $N$ qubits corresponds to a given electronic configuration (Slater determinant) in the entire Fock space. Although the Hilbert space of interest of a given electronic structure problem is usually much smaller than the entire Fock space – i.e., considering a fixed number of electron and spin number – this encoding proves itself very efficient for an interacting system. However, it seems counter-productive for a non-interacting system which can be described by a single Slater determinant. In practice, the non-interacting Hamiltonian is expressed in the mono-particle basis of dimension $N$ for which diagonalization typically scales as $\mathcal{O}(N^3)$, instead of the many-body basis that is infamous for its exponential scaling.

Considering the non-interacting KS Hamiltonian (1) described in finite basis-set $\{\chi_i(\vec{r})\}$ composed of $N$ spin-orbitals, it reads in second quantization:

$$\hat{h}^{\text{KS}} \quad = \quad \sum_{i=1,j=1}^{N} h_{ij}^{\text{KS}} \left( \hat{c}_i^\dagger \hat{c}_j + h.c. \right), \tag{5}$$

where $\hat{c}_i^\dagger$ ($\hat{c}_i$) refers to the creation (annihilation) operator of an electron in the spin-orbital $\chi_i(\vec{r})$, respectively. $h_{ij}^{\text{KS}}$ contains both the kinetic contributions and the KS potential. In the following we propose a transformation to solve this problem with only $M = \mathcal{O}(\log_2 N)$ qubits by mapping $\hat{h}^{\text{KS}}$ of dimension $N \times N$ onto a $M$-qubit system in the spirit of the Harrow–Hassidim–Lloyd algorithm, although the problems to solve are completely different [56]. In complete analogy with the usual encoding of an interacting problem onto a quantum computer, this $M$-qubit system can be interpreted as an auxiliary system of $M$ pseudo spin-orbitals. This means that each state of the computational basis denoted by $\{|I\rangle = |\eta_M^I, \ldots, \eta_1^I\rangle\}$ corresponds to a configuration where each pseudo spin-orbital $\mu$ is either empty ($\eta_\mu^I = 0$) or occupied ($\eta_\mu^I = 1$).

This leads to a direct mapping between the non-interacting Hamiltonian $\hat{h}^{\text{KS}}$ to an auxiliary interacting Hamiltonian $\hat{H}^{\text{aux}}$,

$$\hat{h}^{\text{KS}} \to \hat{H}^{\text{aux}} = \sum_{I=0,J=0}^{2^M-1} H_{IJ}^{\text{aux}} |I\rangle\langle J|, \tag{6}$$

where $I$ and $J$ are the integers corresponding to the binary strings $|I\rangle$ and $|J\rangle$ such that $H_{IJ}^{\text{aux}} = h_{(I+1)(J+1)}$. In other words, each spin-orbital of the KS computational basis-set $|\chi_i\rangle$ is associated to a binary string $|I\rangle$ of the Fock-space of $H^{\text{aux}}$, i.e. a configuration of the auxiliary interacting problem. Note that this mapping is actually arbitrary and that one can map any computational basis state to any orbital. Hence, depending on the architecture of the quantum device, one mapping could be more appropriate than another to reduce the circuit complexity and mitigate the effect of noise.

Let us now introduce the JW transformation for qubit-based-excitation creation (annihilation) operators $\hat{b}_\mu^\dagger$ ($\hat{b}_\mu$) in the auxiliary orbital $\mu$, for $1 \leq \mu \leq M$,

$$\hat{b}_\mu \quad = I^{\otimes \mu-1} \otimes S^- \otimes I^{\otimes M-\mu}, \quad \text{with } S^- = \frac{1}{2}(X + iY), \tag{7}$$

$$\hat{b}_\mu^\dagger \quad = I^{\otimes \mu-1} \otimes S^+ \otimes I^{\otimes M-\mu}, \quad \text{with } S^+ = \frac{1}{2}(X - iY), \tag{8}$$

where $X$, $Y$ and $Z$ are the usual Pauli matrices and $I$ is the identity matrix. Interestingly, the creation and annihilation operators of our fictitious interacting system do not have to fulfil

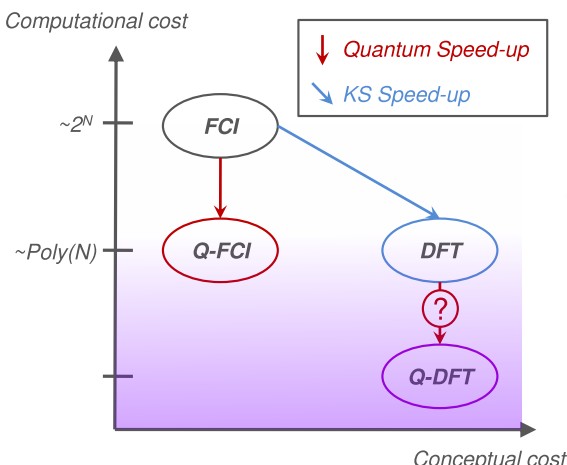

Figure 1: Computational cost to extract the ground-state energy for full configuration interaction (FCI) and DFT, with or without the use of quantum computers. Q-FCI and Q-DFT refer to the analog of FCI and DFT on quantum computers, respectively.

anticommutation rules, in contrast to the original system composed of fermions. Hence, the JW transformation leads to local operations without the usual large string of $Z$ Pauli matrices, which leads to non-local operations. Note that other transformations such as Bravyi–Kitaev could also be used [55]. The projector in Eq. (6) can be written in terms of these qubit-based excitation operators as follows:

$$|I\rangle\langle J| = \prod_{\mu=1}^{M} \left(\hat{b}_\mu \hat{b}_\mu^\dagger\right)^{(1-\eta_\mu^I)(1-\eta_\mu^J)} \left(\hat{b}_\mu\right)^{(1-\eta_\mu^I)\eta_\mu^J} \left(\hat{b}_\mu^\dagger\right)^{\eta_\mu^I(1-\eta_\mu^J)} \left(\hat{b}_\mu^\dagger \hat{b}_\mu\right)^{\eta_\mu^I \eta_\mu^J} . \tag{9}$$

The Hamiltonian in Eq. (6) is now written as a linear combination of Pauli strings, which expectation values can be measured on a quantum computer. As readily seen in Eqs. (3) and (4), the occupied KS orbitals and associated energies are required to compute the electronic density and the ground-state energy, respectively. At each iteration of the KS self-consistent procedure, they are solutions of the Schrödinger equation

$$\hat{H}^{\text{aux}} |\varphi_k\rangle = \varepsilon_k |\varphi_k\rangle , \tag{10}$$

for which quantum computers are originally hoped to provide an exponential speed-up compared to classical computers.

In Fig. 1, we represent the possible improvement in computational cost by using the KSDFT formalism together with quantum computers. The full configuration interaction (FCI) method – equivalent to exact diagonalization – scales exponentially with respect to the system size, and an exponential speed-up is given by the KSDFT formalism or by employing quantum computers (Q-FCI). If KSDFT can also benefit from quantum computers (Q-DFT), one could scale up the whole range of applications of quantum chemistry. Note that each method in Fig. 1 is in principle exact, although approximate functionals are used in practice in DFT.

## 2.2 Kohn–Sham self-consistent conditions on a quantum device

Given the KS equation mapped onto an auxiliary interacting Hamiltonian in Eq. (10), we propose to use a NISQ-adapted variational quantum eigensolver (VQE) [19, 20] to efficiently extract the occupied KS orbitals energies on quantum computers. Because in our context, one has to solve Eq. (10) for the first $N_{\text{occ}}$ lowest-energy states, we naturally turn to the ensemble

extension of VQE (ensemble-VQE) [25, 26]. Within ensemble-VQE, we consider an ensemble of $N_{\text{occ}}$ states,

$$|\varphi_k(\boldsymbol{\theta})\rangle = \hat{\mathcal{U}}(\boldsymbol{\theta})|\phi_k\rangle, \quad 1 \leq k \leq N_{\text{occ}}, \tag{11}$$

where $\{|\phi_k\rangle\}$ is an orthonormal basis issued from the computational basis and simple to prepare. $\hat{\mathcal{U}}(\boldsymbol{\theta})$ is usually referred to the circuit ansatz, i.e. a unitary transformation composed of $\boldsymbol{\theta}$-parametrized quantum gates. The set of parameters $\boldsymbol{\theta}$ are classicaly optimized by minimizing the ensemble energy,

$$E^{\text{ens}} = \min_{\boldsymbol{\theta}} \left\{ \sum_{k=1}^{N_{\text{occ}}} w_k \langle \varphi_k(\boldsymbol{\theta})| \hat{H}^{\text{aux}} |\varphi_k(\boldsymbol{\theta})\rangle \right\}. \tag{12}$$

Considering $w_1 > \ldots > w_k > \ldots > w_{N_{\text{occ}}}$, it results that the optimized states are by construction approaching the KS orbitals, $|\varphi_k(\boldsymbol{\theta}^*)\rangle \simeq |\varphi_k\rangle$ [25] (up to the error of the classical optimizer and the expressibility of the ansatz), where $\boldsymbol{\theta}^*$ are the minimizing parameters in Eq. (12). The first energy term in the right-hand side of Eq. (4) then reads

$$\mathcal{E}^{\text{KS}} \approx \mathcal{E}^{\text{KS}}(\boldsymbol{\theta}^*) = \sum_{k=1}^{N_{\text{occ}}} \langle \varphi_k(\boldsymbol{\theta}^*)| \hat{H}^{\text{aux}} |\varphi_k(\boldsymbol{\theta}^*)\rangle. \tag{13}$$

Note that in contrast to the usual WFT problems studied within VQE, the auxiliary many-body problem described in this paper is not physical anymore, and no constraints on the spin or the number of particles have to be considered. In other words, the entire space spanned by the $M$ qubits corresponds to the space spanned by the $N = 2^M$ KS orbitals. Consequently, a hardware efficient ansatz can be used for $\hat{\mathcal{U}}(\boldsymbol{\theta})$ instead of physically motivated ansatz such as the unitary coupled cluster ansatz which requires very deep circuits.

At this stage it is important to highlight that we have a direct access to the occupation number of orbitals in the computational basis. Indeed, repeated measurements of the state $|\varphi_k\rangle = \sum_I \varphi_k(I)|I\rangle$ give access to $|\varphi_k(I)|^2$, the probability to measure $|I\rangle$, and thus to the computational basis orbitals occupations

$$n_I = \sum_{k=1}^{N_{\text{occ}}} |\varphi_k(I)|^2. \tag{14}$$

Within the lattice version of DFT, known as *Site-Occupation Functional Theory* (SOFT) [57, 58], the orbital dependent XC potential $\mathbf{v}_{\text{xc}}[\mathbf{n}]$, that only depends on orbital occupations $\mathbf{n} = \{n_1, \ldots, n_I, \ldots, n_N\}$, is then straightforwardly accessible to loop the KS protocol.

Going back to standard DFT, the knowledge of the computational basis-set occupation numbers is not sufficient to access the real-space electronic density in Eq. (3). Indeed, inserting closure relations in Eq. (3) leads to

$$
\begin{aligned}
\mathbf{n}(\vec{r}) &= \sum_{i=1,j=1}^{N} \sum_{k=1}^{N_{\text{occ}}} \langle \vec{r}|\chi_i\rangle \langle \chi_i|\varphi_k\rangle \langle \varphi_k|\chi_j\rangle \langle \chi_j|\vec{r}\rangle \\
&= \sum_{i=1,j=1}^{N} \langle \vec{r}|\chi_i\rangle \langle \chi_j|\vec{r}\rangle \sum_{k=1}^{N_{\text{occ}}} \varphi_k(I)\varphi_k(J),
\end{aligned} \tag{15}
$$

where we recall that the original KS computational basis set $\{|\chi_i\rangle\}$ is orthonormal and is translated to quantum computers by the mapping $|\chi_i\rangle \rightarrow |I\rangle$. Hence, inferring the real-space density

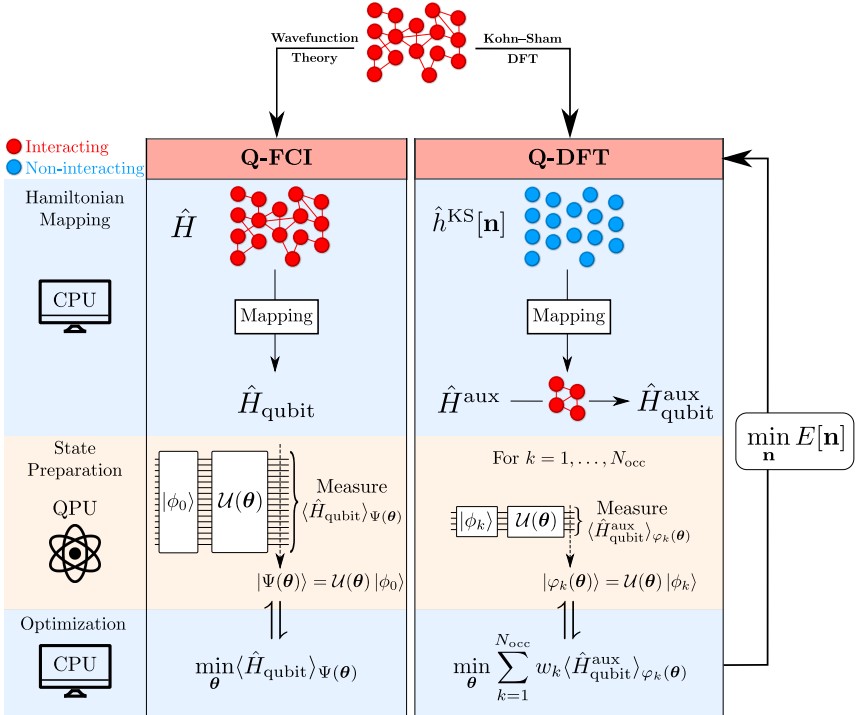

Figure 2: Flowchart of the quantum algorithm analog of FCI (Q-FCI) and DFT (Q-DFT) for $N = 16$ spin-orbitals, using variational quantum algorithms as solvers. The ground-state VQE and ensemble-VQE are used in Q-FCI and Q-DFT, respectively.

requires the ability to extract the product between different amplitudes $\varphi_k(I)$ and $\varphi_k(J)$. Interestingly, this product can be estimated directly by repeated measurements of the operator

$$\hat{\Gamma}_{IJ}^{\text{aux}} = |I\rangle \langle J| + |J\rangle \langle I| , \tag{16}$$

after transforming $\hat{\Gamma}_{IJ}^{\text{aux}}$ into a linear combination of Pauli strings for instance, as follows

$$\langle \hat{\Gamma}_{IJ}^{\text{aux}} \rangle_{\varphi_k(\boldsymbol{\theta})} = 2\text{Re}\left(\varphi_k(I)\varphi_k(J)\right) . \tag{17}$$

Note that non-relativistic quantum chemistry has real algebra and real-algebra ansatz can be used to ensure that orbital coefficients remain real, so that the extraction of the imaginary part is not needed. Also, most of the expectation values of Pauli strings required to estimate each element in Eq. (17) are already measured when performing the ensemble-VQE in Eq. (12). The flowchart of the Q-DFT algorithm is depicted in Fig. 2.

## 3 Results and Discussions

### 3.1 Methods

After the transformation of the non-interacting Hamiltonian into the auxiliary interacting one [see Eqs. (6) and (9)], we classically simulate the ensemble-VQE algorithm [25, 26] using the Qiskit package [59] and a state-vector simulation (without noise). The hardware efficient ansatz, the $R_y$ ansatz, is used for $\hat{\mathcal{U}}(\boldsymbol{\theta})$ in Eq. (11), and is given by [60]

$$\hat{\mathcal{U}}(\boldsymbol{\theta}) = \prod_{m=1}^{M} R_{y,m}(\theta_{m_y}^0) \prod_{n=1}^{N_L} \hat{\mathcal{U}}_n^{\text{ENT}}(\boldsymbol{\theta}^n), \tag{18}$$

$q_0:$ — $R_Y(\theta_0)$ — • — $R_Y(\theta_3)$ — • — $R_Y(\theta_6)$ — • — $R_Y(\theta_9)$ — • — $R_Y(\theta_{12})$ —

$q_1:$ — $R_Y(\theta_1)$ — ⊕ — • — $R_Y(\theta_4)$ — ⊕ — • — $R_Y(\theta_7)$ — ⊕ — • — $R_Y(\theta_{10})$ — ⊕ — • — $R_Y(\theta_{13})$ —

$q_2:$ — $R_Y(\theta_2)$ — ⊕ — $R_Y(\theta_5)$ — ⊕ — $R_Y(\theta_8)$ — ⊕ — $R_Y(\theta_{11})$ — ⊕ — $R_Y(\theta_{14})$ —

Figure 3: Illustration of the $R_y$ hardware efficient ansatz of Eqs. (18) and (19) with $N_L = 4$.

for a number of layers $N_L$ and a number of qubits $M$. The entanglement unitary blocks read

$$\hat{\mathcal{U}}_n^{\mathrm{ENT}}(\boldsymbol{\theta}^n) = \prod_{m=1}^{M-1} \mathrm{CNOT}_{m(m+1)} \prod_{m=1}^{M} R_{y,m}(\theta_m^n). \tag{19}$$

As a proof a concept, we study the one-dimensional Hubbard model with 8 sites and $N_e = 4$ electrons, and one chain of 8 hydrogens in the minimal STO-3G basis (i.e., $N_e = 8$ electrons in 8 spatial-orbitals), using SOFT and regular DFT, respectively. Their implementation on quantum computers is denoted as Q-SOFT and Q-DFT, respectively. As our algorithm is designed for an orthonormal basis set, we used the Löwdin symmetric orthonormalization of the atomic orbitals to get a basis of orthonormal atomic orbitals. In principle, this step will be circumvented by using already orthonormal basis sets such as plane waves or Daubechies wavelets [61, 62]. They usually require much more basis functions but this is not a problem for Q-DFT as they are mapped on only $\log_2(N)$ qubits. Note that both systems are composed of 16 spin-orbitals and they should in principle be described by 16 qubits with the Jordan–Wigner transformation. As they are closed-shell systems and that no relativistic effects are considered, we can actually select only one specific block of the KS Hamiltonian, i.e., either the spin-$\alpha$ or spin-$\beta$ block of $N = 8$ spin-orbitals, such that the number of qubits required by our quantum algorithm is equal to $M = \log_2(8) = 3$ only. To solve the first $N_{\mathrm{occ}} = N_e/2$ states (corresponding to the first KS occupied orbitals), we use the ensemble-VQE solver for an ensemble of $N_{\mathrm{occ}}$ states, with ensemble weights defined as $w_k = (1 + N_{\mathrm{occ}} - k)/(N_{\mathrm{occ}}(N_{\mathrm{occ}} + 1)/2)$ such that $w_{k+1} < w_k$ and $\sum_{k=1}^{N_{\mathrm{occ}}} w_k = 1$. The initial states in Eq. (11) are taken to be the first states in the computational basis, i.e. $\{|\phi_k\rangle\} = \{|000\rangle, |001\rangle\}$ for the Hubbard model, and $\{|\phi_k\rangle\} = \{|000\rangle, |001\rangle, |010\rangle, |011\rangle\}$ for the hydrogen chain.

The $R_y$ ansatz with $N_L = 4$ layers is used, thus leading to $M(N_L + 1) = 15$ ansatz-parameters, as illustrated in Fig. 3. For the noiseless state-vector simulation, these parameters are optimized classically with the L-BFGS-B optimizer, whereas the simultaneous perturbation stochastic approximation (SPSA) optimizer with a maximum of 5000 iterations is used when sampling noise is considered. Sampling noise is simulated by drawing samples from a multinomial distribution of the states. We considered three distributions with a total number of shots per energy evaluation of $10^6$, $10^5$ and $10^4$ shots, which are equally shared between the number of Pauli strings composing the KS Hamiltonian (9 for the Hubbard model and 20 for the hydrogen chain), such that the actual number of shots per Pauli is around one order of magnitude below the total number of shots. In practice, sampling noise doesn't allow to have the same convergence criteria for the self-consistent KS equations than in noiseless state-vector simulations. Hence, the convergence was stopped manually by considering a maximum number of 20 iterations for the outer KS self-consistent loop. Finally, the Bethe ansatz local density approximation (BALDA) is used to model the Hxc potential and Hxc energy functional for the Hubbard model [63], and the local spin density functional SVWN for the hydrogen chain.

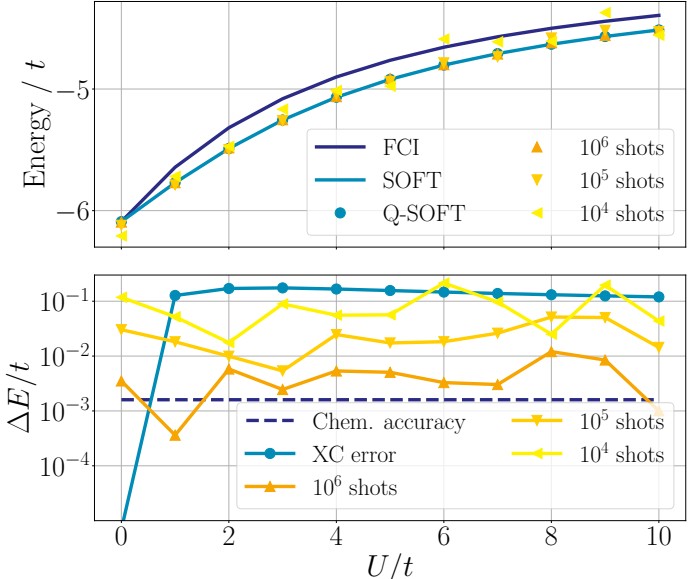

Figure 4: **Top panel:** Ground-state energy of the inhomogeneous one-dimensional Hubbard model as a function of the interaction strength $U/t$, in units of $t$. **Bottom panel:** Absolute error between Q-SOFT and SOFT energies when sampling noise is considered. They are compared to the chemical accuracy and the XC-error.

## 3.2 Solving the Kohn–Sham self-consistent equations

Starting with the one-dimensional inhomogeneous Hubbard model, the Hamiltonian reads

$$\hat{H} = -t \sum_{i=1}^{N} \sum_{\sigma} \left( \hat{c}_{i\sigma}^{\dagger} \hat{c}_{(i+1)\sigma} + \hat{c}_{(i+1)\sigma}^{\dagger} \hat{c}_{i\sigma} \right) + U \sum_{i=1}^{N} \hat{n}_{i\uparrow} \hat{n}_{i\downarrow} + \sum_{i=1}^{N} v_i \hat{n}_i \,,$$

(20)

where $\hat{n}_i = \hat{n}_{i\uparrow} + \hat{n}_{i\downarrow}$ is the occupation number operator, with $\hat{n}_{i\sigma} = \hat{c}_{i\sigma}^{\dagger} \hat{c}_{i\sigma}$, and $t$ and $U$ are the hopping integral and the on-site electron-electron interaction, respectively. The external potential is taken as $v_i = (i-1)/10$ and antiperiodic boundary conditions ($\hat{c}_{(N+1)\sigma} = -\hat{c}_{1\sigma}$) are used.

The total ground-state energy calculated from Eq. (4) is shown on the top panel of Fig. 4, using noiseless state-vector simulation (denoted by Q-SOFT) and simulations with sampling noise. As readily seen in Fig. 4, the noiseless state-vector simulation is on top of the SOFT energy, with an error inferior to $10^{-5}$ units of $t$, thus demonstrating the capability of the $R_y$ ansatz to capture the orbital energies and occupation numbers with sufficient accuracy. Adding sampling noise doesn't have a significant impact for $10^6$ and $10^5$ shots, as the resulting ground-state energies doesn't deviate much from the noiseless Q-SOFT energy. However, considering $10^4$ shots only leads to non-negligible errors.

Let us turn to the quantum simulation of regular DFT of the $H_8$ linear chain. On the top panel of Fig. 5, we compare the ground-state energies obtained by the noiseless Q-DFT simulation with the ones obtained by regular DFT (on classical computer) and FCI, performed with Psi4 [64] in the same basis. As for the Hubbard model, the noiseless Q-DFT simulation using the L-BFGS-B optimizer is in excellent agreement with DFT, with an error of about $10^{-4}$ hartree, meaning that the $R_y$ ansatz with 4 layers is also sufficient in this case. In constrast to Q-SOFT where only the orbital occupation have to be evaluated [see Eq. (14)], one has to compute the real-space electronic density given by Eq. (3) by evaluating the expectation value in Eq. (17). In the case of $H_8$, it requires the estimation of the expectation value of 36 Pauli

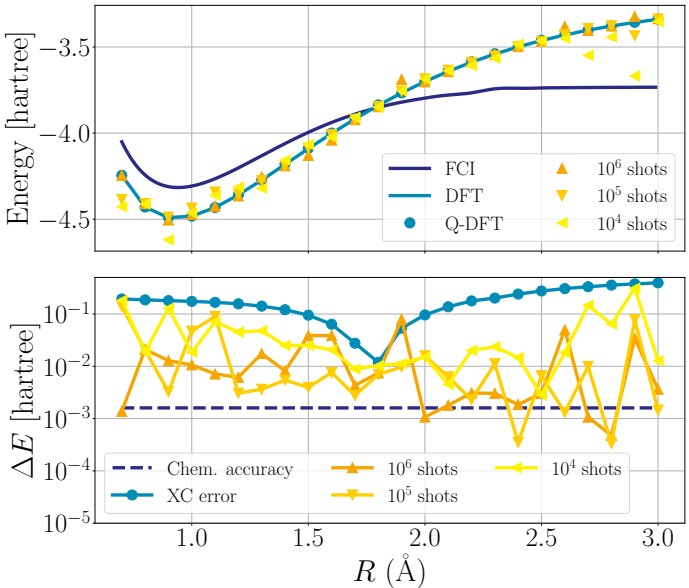

Figure 5: **Top panel:** Ground-state energy of the $H_8$ linear chain as a function of the interatomic distance, in hartree. **Bottom panel:** Absolute error between Q-DFT and DFT energies when sampling noise is considered. They are compared to the chemical accuracy and the XC error.

strings (among which 20 are already required to evaluate the KS orbital energies). Globally, a good agreement is also obtained even with sampling noise, except for few interatomic distances. As these distances are not the same when considering different number of shots, it means that the error is not tied to the system itself but comes from other sources: the stochastic optimization of the ensemble-VQE by SPSA together with the fact that the KS self-consistent equations are not strictly converged. This could be potentially fixed by considering more KS iterations and by averaging the last ten or twenty iterations, as commonly done with SPSA.

Let us now turn to the bottom panels of Figs. 4 and 5, featuring the absolute error between the quantum simulations and the classical reference, in comparison to the chemical accuracy and the XC error (comprising the density-driven and functional-driven error). Until now, quantum computers have been promised to help solving the many-body problem with chemical accuracy (1.6 milliHartree). This goal is extremely challenging, to the point that quantum supremacy for chemistry might only be attainable in several decades. On the other hand, DFT is itself subject to approximations coming from the XC functional. Hence, the accuracy that Q-SOFT and Q-DFT have to reach is bounded by this XC error. Note also that the bigger the system is, the larger the error between the DFT and FCI total energy will be. As seen by the logarithmic scale on the top panel of Fig. 4, the absolute error between Q-SOFT with sampling noise and SOFT is in between $10^{-1}$ and $10^{-2}$ units of $t$ for $10^5$ shots, and in between $10^{-2}$ and $10^{-3}$ units of $t$ for $10^6$ shots. In comparison, the XC error is higher than $10^{-1}$ units of $t$ everywhere except at $U = 0$ for which the BALDA is exact in the thermodynamic limit, and the sampling noise error with $10^4$ shots is generally just slightly below. This XC error allows to bypass the burden of reaching chemical accuracy, such that Q-SOFT appears much more achievable with noisy devices than quantum algorithms for wavefunction theory. The same analysis can be drawn for $H_8$, on the top panel of Fig. 5, except that there is no significant difference between the simulations with sampling noise when different number of shots is considered, though considering $10^4$ shots only seems to deteriorate the results slightly.

## 3.3 Discussion on the numerical efficiency

The validity and applicability of the concept being proven, the "numerical" efficiency and potential computational gain should be discussed.

In SOFT, the convergence of the self-consistent KS equations requires the calculation of the orbital occupations [Eq. (14)], while regular DFT requires the real-space density [Eq. (15)]. Both necessitate the evaluation of product of amplitudes $\varphi_k(I)\varphi_k(J)$ ($I = J$ for the orbital occupations) obtained by measuring the elements of the density matrix operator [see Eq. (17)]. The number of elements scales exponentially with the number of qubits and quadratically with the number of orbitals of the original system. As there are $N_{\text{occ}} \propto N_e$ states, the overall scaling to measure the orbital occupations is $\mathcal{O}(2^M N_e) = \mathcal{O}(N N_e)$, and $\mathcal{O}(N^2 N_e)$ for the real-space density and the orbital energies. Owing the sparse nature of the KS Hamiltonian [65], this estimation represents a largely overestimated bound, as most of the $\varphi_k(I)\varphi_k(J)$ elements are equal to zero. Note also that there might be more efficient representations than a linear combination of Pauli string in terms of number of terms in the Hamiltonian, using for instance partition strategies [20, 66].

In addition, one has to consider the number of measurements and the number of iterations performed in VQE. As shown in Ref. [67], the number of measurements to achieve a given accuracy $\epsilon$ in estimating the energy scales with the square of the one-norm of the Hamiltonian. For the electronic many-body problem, the one-norm typically scales in between $\mathcal{O}(N^2)$ and $\mathcal{O}(N^3)$, but several techniques can be used to reduce this scaling close to $\mathcal{O}(N)$ such as double factorization [32], tensor hypercontraction [68], $n$-representability constraints [69], and rotations of the orbital basis [70]. Turning to the KS Hamiltonian – or the auxiliary Hamiltonian in Eq. (6) – the number of interacting pseudo-orbitals is only $\log_2(N)$, and we expect its one-norm to scale sublinearly with $N$. A more detailed study of the one-norm of the Q-DFT Hamiltonian for large systems would be required to confirm our assumption, and is left for future work.

Turning to the number of iterations, it has been shown that optimizing the circuit parameters in any variational quantum algorithm is NP-hard [71], such that we should not expect any advantage in using ensemble-VQE as a solver. However, in practice with an approximate ansatz and a finite maximal number of iterations, we can reach approximate states and orbital energies with sufficient accuracy for the method to be valuable, in analogy with VQE applied to the many-body ground-state problem. In Q-DFT, chemical accuracy is not always desired and an error of around one-order of magnitude below the XC error is acceptable. Still, using other non-variational solvers such as the quantum phase estimation and the iterative phase estimation algorithm is currently under investigation.

Although these scalings question a practical quantum advantage for SOFT or regular DFT, they are still below the classical upper-bound of $\mathcal{O}(N^3)$, and all the measurements of all the terms to compute the expectation values are completely independent from each other and straightforwardly parallelisable on multiple quantum computers. Within this strong assumption, these operations could be omitted in the estimation of the complexity, and the limiting step to implement SOFT and DFT on a quantum computer would be dictated by the gate complexity. Alternatively, the $\mathcal{O}(4^M) = \mathcal{O}(N^2)$ expectation values can actually be approximated with only $poly(M) = poly(\log_2(N))$ measurements of our prepared states $\{|\varphi_k\rangle\}$ using shadow tomography [72].

Let us now focus on the gate complexity, that is related to the number of orbitals and the ansatz operator in the ensemble-VQE simulation. Considering a system of $M$ interacting orbitals, the different variants of the unitary coupled-cluster ansatz (truncated to single and double excitations) would typically scale as $\mathcal{O}(M^4)$. Instead, hardware efficient ansatz are sufficient to treat our auxiliary interacting system, which features shallower circuit at the expense of more variational parameters given to the classical optimizer. The $R_y$ ansatz features

$N_L(M-1)$ two-qubit gates, thus leading to a gate complexity that scales as $\mathcal{O}(M^2)$, assuming that the number of layers $N_L$ grows linearly with $M$ (checking this assumption is left for future work). Therefore, the gate complexity required to diagonalize the KS Hamiltonian on quantum computers is about $\mathcal{O}(M^2) = \mathcal{O}(\log_2(N)^2)$ using hardware efficient ansatz within ensemble-VQE. In comparison with the $\mathcal{O}(N^3)$ upper bound on classical computers, a substantial speed-up can yet be envisioned.

As mentioned at the beginning of the section, self-consistency requires the orbital occupations [Eq. (14)] within SOFT, while for regular DFT the real-space density [see Eq. (15)] needs to be reconstructed on the classical computer, which required $\mathcal{O}(N^2 N_e)$ operations. Hence, SOFT appears optimal when applied to quantum computers, however it remains mainly applied to lattice Hamiltonians and model systems, even though recent promising works aim to extend it to real quantum chemistry problems [73, 74]. Nevertheless, a quantum advantage can still be envisioned as the determination of the KS orbitals and energies scales as $\mathcal{O}(M^2) = \mathcal{O}(\log_2(N)^2)$ instead of the classical upper bound of $\mathcal{O}(N^3)$ (assuming quantum computer parallelization). It should also be noted that in quantum chemistry, a polynomial speed-up in the computational cost is already considered as a significant improvement [34]. All together, Q-DFT could potentially lead to an advantage over DFT on a classical computer.

Finally, in this paper we focused on the advantage of solving the KS self-consistent equations. In some cases, the rate determining-step is actually the construction of the KS Hamiltonian itself, and more specifically the electron repulsion integrals to form the Coulomb matrix. There has been numerous studies to reduce the computational cost of this construction (see Ref. [75] and references therein). For large molecules, diagonalization remains the major obstacle for DFT calculations [75].

# 4 Conclusions

In this work, we show that mean-field-type approaches such as DFT are also expected to benefit from the advance of quantum technologies, opening the door for quantum simulations of systems having millions of orbitals. As a proof of concept, we simulated the one dimensional Hubbard model with a quantum algorithm implementation of site occupation functional theory, the lattice version of DFT, for which an exponential speed-up can be envisioned, assuming that parallel quantum computing is accessible or using shadow tomography [72]. Then, the quantum analog of regular DFT was successfully applied to a hydrogen chain. Though an exponential speed-up is not expected due to the reconstruction of the real-space density, solving the KS self-consistent equation on a quantum device might still lead to a quantum advantage. Moreover, a significant advantage of performing DFT on quantum computer compared to WFT is the level of accuracy that one needs to achieve. Because, WFT targets chemical accuracy, a practical quantum advantage for WFT and quantum chemistry in the NISQ era is currently surrounded by skepticism [34]. However, DFT has its own level of approximations coming from the approximate XC functional. This error being orders of magnitude larger than chemical accuracy – except in cases with significant error cancellation in the estimation of energy differences – we can expect DFT on quantum computers to be much more resilient to noise than WFT and to be the nearest-term application of quantum computer for not only quantum chemistry, but also condensed-matter physic and even biology.

**Author contributions** All the authors contributed equally. M. S. and B. S. designed the mapping, S. Y. and B. S. implemented the algorithm, and S. Y., M. S. and B. S. wrote the manuscript.

**Funding information** This work has benefited from a state grant managed by the French National Research Agency under the Investments of the Future Program with the reference ANR-21-ESRE-0032.

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
