# Peer review of "Toward Density Functional Theory on Quantum Computers?"

_SciPost Physics, doi:SciPost Phys. 14, 055 (2023)_

## Round 2 · Referee Report · Anonymous (Referee 1) · 2022-6-23

Strengths

  1. Well written
  2. Well organized
  3. Interesting research
  4. Opens the door for many follow up studies (on devices, applications, theoretical extensions, etc.)

Weaknesses

  1. Slightly under cited - missing a few key references
  2. A few technical and figure clarifications are needed

Report

In this manuscript, the authors propose an algorithm to perform density functional theory electronic structure calculations on a quantum computer. They demonstrate this algorithm on an ideal state vector simulator, as well as IBM’s regular simulator on the 1D Hubbard model and on an 8 atom hydrogen chain. Their results are in good agreement with classical calculations. The research itself is interesting, the paper is very well written, and I believe it should be published with a few adjustments.

Larger comments: - There are a few missing references to fairly related works — - Highlighting what is done in the present manuscript as compared to this manuscript would be important: https://journals.aps.org/prb/abstract/10.1103/PhysRevB.79.205117 - https://www.nature.com/articles/srep00391 - https://aip.scitation.org/doi/10.1063/5.0029536 and https://www.nature.com/articles/s41524-020-00353-z (neither are directly related — DFT is the classical low level solver, but worth including and checking other recent works out of these groups) - https://iopscience.iop.org/article/10.1088/2058-9565/ac1ca6 (A good review that would help motivate the current manuscript) - https://arxiv.org/ftp/arxiv/papers/1903/1903.05550.pdf - Generally one should be careful with the discussion of quantum advantage, as it has been shown by a few groups for a few tasks — the challenge is a practical quantum advantage or a useful quantum advantage. Generally the authors make this distinction well but it’s worth double checking in a few places (one newer claim came out this month that could be worth citing: https://www.nature.com/articles/s41586-022-04725-x) - Same goes for the discussion of chemical accuracy — I believe the team at google did obtain chemical accuracy for an H8 chain in the Hartree-Fock Science paper — they just used fairly expensive classical mitigation techniques to get there. The statement “Quest for chemical precision remains a long journey” while accurate, might want to be written with a little bit more nuance given the literature that exists - Some of the algorithm discussion would be greatly strengthened by a diagram demonstrating how the algorithm works, which pieces are done on a classical device versus a quantum device, etc. - In the data for both Figures 3 and 4, it looks like 10^5 and 10^6 shots produce roughly the same results. What happens when you use fewer? A big sell of this algorithm could be the lack of dependency on the number of shots required (to an extent), and it would be interesting to know what the minimum amount required to obtain these results would be - Did the authors try using a noisy simulator? I.e. loading the noise profile of one of the devices in the QASM simulation? The majority of present algorithms aren’t hindered by sampling noise but by device noise, so it would be interesting to see how this algorithm would do either with a noisy simulator or on a real device because that tends to be the prohibitive step - A little more detail on the ensemble VQE method might be useful (could also be clarified with an algorithm diagram)

A few minor comments and edits: - There’s an overhang in the in-line equation right after Eq. 1 - While all the acronyms are popularly used, the paper is generally so well written that sometimes the acronyms make it more challenging to follow. It might be worth minimizing acronym use throughout if it doesn’t feel too repetitive - In a few equations, “+h.c.” is used, which is again common practice, but actually including the hermitian conjugate term only adds one term and it would be worth including for completion (Eq. 18 and in-text equation right above Eq. 15) - Typo: “12 qubit plateform” -> “12 qubit platform” - Typo: “at the opposite of what has been done for decades.” —> “as opposed to what has been done for decades.”? - Figure 1: - I don’t think the authors discuss the meaning of the colored bands in the background. It goes from white, to blue, to pink which presumably represents different scaling ranging, but what are these ranges? Or is this coloring stylistic? - Along these lines, I don’t think the molecules on the right of Figure 1 are discussed — are these corresponding to what can be accomplished in each scaling regime? If so, it’s a little bit deceiving because while something like FeMoco might be a size that can be considered with DFT, the calculation wouldn’t be accurate due to the strong correlation known to exist within the complex. It might be worth putting a material or something where DFT is known to succeed (and wavefunction or density matrix based methods can’t perform) - Figure 4: The FCI calculation for H8 is cheap enough that it might be worth sampling a little bit more frequently such that the curve in the top panel is smooth (a few edges appear near the equilibrium distance and around 2.25\AA) - On page 9 under Figure 3 there is the statement “For now on..” —> maybe meant to be “From now on…” - Typo in the first line of Section 3.3 “The validity and applicability of the concept being proved” —> proven instead of proved

Requested changes

  1. The addition of a few key references as mentioned above
  2. A distinction between the present work and the PRB added above
  3. A few typo/language changes
  4. Clarification of parts of Figure 1
  5. Some language tweaks for the discussions of quantum advantage and chemical accuracy
  6. Potentially an algorithm overview figure

  • validity: good
  • significance: good
  • originality: good
  • clarity: high
  • formatting: good
  • grammar: excellent

Author:  Matthieu Saubanere  on 2022-07-04  [id 2629]

(in reply to Report 1 on 2022-06-23)
Category:
answer to question
reply to objection

The referee writes:

In this manuscript, the authors propose an algorithm to perform density functional theory electronic structure calculations on a quantum computer. They demonstrate this algorithm on an ideal state vector simulator, as well as IBM's regular simulator on the 1D Hubbard model and on an 8 atom hydrogen chain. Their results are in good agreement with classical calculations. The research itself is interesting, the paper is very well written, and I believe it should be published with a few adjustments.

Our response: We thank the referee for his/her careful reading of the manuscript and his interest in our work. We answer to his/her suggestions below.

The referee writes:

Larger comments: There are a few missing references to fairly related works: - Highlighting what is done in the present manuscript as compared to this manuscript would be important: - Ref.1 https://journals.aps.org/prb/abstract/10.1103/PhysRevB.79.205117) - Ref.2 https://www.nature.com/articles/srep00391 - Ref.3 https://aip.scitation.org/doi/10.1063/5.0029536 and https://www.nature.com/articles/s41524-020-00353-z (neither are directly related -- DFT is the classical low level solver, but worth including and checking other recent works out of these groups) - Ref.4 https://iopscience.iop.org/article/10.1088/2058-9565/ac1ca6 (A good review that would help motivate the current manuscript) - Ref.5 https://arxiv.org/ftp/arxiv/papers/1903/1903.05550.pdf

Our response: We thank the referee for these additional references. We agree to add references [3-5] to strengthen our introductions. As mentioned by the referee, Refs. [1-2] are indeed very interesting and we agree to discuss their differences with our work. In fact both Ref. [1-2] use DFT, originally developed for the electronic Hamiltonian, as a tool applied to quantum computing problems. More precisely they demonstrate that both Hohenberg--Kohn and Runge--Gross theorems also hold for qubit Hamiltonians associated to quantum computing challenges (e.g. to the NP-Complete MAXCUT problem in Ref. [1]). In contrast, our work aim to use quantum computing as a tool to solve the "original electronic'' DFT equations associated to the original electronic Hamiltonian used in the context of quantum chemistry/condensed matter physics. To that aim, we propose to map the "original electronic" $N$ orbitals Kohn--Sham non-interacting problem onto an interacting problem described by a qubit Hamiltonian with $M = log(N)$ qubits, paving the way for an efficient resolution of the "original electronic" KSDFT equation on quantum computer.

The referee writes:

Generally one should be careful with the discussion of quantum advantage, as it has been shown by a few groups for a few tasks -- the challenge is a practical quantum advantage or a useful quantum advantage. Generally the authors make this distinction well but it's worth double checking in a few places (one newer claim came out this month that could be worth citing: Ref. 7 https://www.nature.com/articles/s41586-022-04725-x

Our response: We agree that discussion of quantum advantage should be taken very carefully, and talking about practical or useful quantum advantage would indeed avoid any confusion. We will change the manuscript accordingly.

The referee writes:

Same goes for the discussion of chemical accuracy -- I believe the team at google did obtain chemical accuracy for an H8 chain in the Hartree-Fock Science paper -- they just used fairly expensive classical mitigation techniques to get there. The statement `Quest for chemical precision remains a long journey' while accurate, might want to be written with a little bit more nuance given the literature that exists

Our response: We understand the comment of the referee. Indeed, for very short circuits as in the Hartree--Fock experiment and in other applications using hardware efficient ansatz with a small number of qubits and layers, chemical accuracy can be achieved with expensive classical mitigation techniques. However, for the size and complexity of systems of current interest, it remains a long journey. Hence, we will modify the sentence as follows: "Quest for chemical precision remains a long journey for relevant applications of high societal and industrial impact".

The referee writes:

Some of the algorithm discussion would be greatly strengthened by a diagram demonstrating how the algorithm works, which pieces are done on a classical device versus a quantum device, etc.

Our response: We thank the referee for his/her suggestion and we will add a diagram summarizing how Q-DFT works together with ensemble-VQE.

The referee writes:

In the data for both Figures 3 and 4, it looks like $10^5$ and $10^6$ shots produce roughly the same results. What happens when you use fewer? A big sell of this algorithm could be the lack of dependency on the number of shots required (to an extent), and it would be interesting to know what the minimum amount required to obtain these results would be

Our response: We agree for Figure 4 (H$_8$) but not for Figure 3 (Hubbard model) where we see a clear difference of about one order of magnitude between $10^5$ and $10^6$ shots. We propose to add different sampling noise for H$_8$ in a revised version with lower number of shots, e.g. $10^4$.

The referee writes:

Did the authors try using a noisy simulator? I.e. loading the noise profile of one of the devices in the QASM simulation? The majority of present algorithms aren't hindered by sampling noise but by device noise, so it would be interesting to see how this algorithm would do either with a noisy simulator or on a real device because that tends to be the prohibitive step.

Our response: We did consider it and did some tests on a smaller system (H$_4$, i.e. two qubits, two states, and one layer) using the noisy classical-simulation proposed in qiskit. Preliminary results were promising, but we plan to investigate this further with more details in a future work, certainly on larger systems with error mitigation techniques, and on a real quantum computer.

The referee writes:

A little more detail on the ensemble VQE method might be useful (could also be clarified with an algorithm diagram)

Our response: We plan to add an algorithm diagram for Q-DFT as suggested by the referee, and ensemble-VQE will be detailed in this same diagram.

The referee writes:

A few minor comments and edits: - There's an overhang in the in-line equation right after Eq. 1 - While all the acronyms are popularly used, the paper is generally so well written that sometimes the acronyms make it more challenging to follow. It might be worth minimizing acronym use throughout if it doesn't feel too repetitive - In a few equations, "+h.c." is used, which is again common practice, but actually including the hermitian conjugate term only adds one term and it would be worth including for completion (Eq. 18 and in-text equation right above Eq. 15) - Typo: "12 qubit plateform" $\rightarrow$ "12 qubit platform" - Typo: "at the opposite of what has been done for decades."$\rightarrow$ " as opposed to what has been done for decades?

Our response: We thank the referee for pointing these typos out, and we will correct them.

The referee writes:

Figure 1: - I don't think the authors discuss the meaning of the colored bands in the background. It goes from white, to blue, to pink which presumably represents different scaling ranging, but what are these ranges? Or is this coloring stylistic?

Our response: This coloring was indeed stylistic.

The referee writes:

Along these lines, I don't think the molecules on the right of Figure 1 are discussed -- are these corresponding to what can be accomplished in each scaling regime? If so, it's a little bit deceiving because while something like FeMoco might be a size that can be considered with DFT, the calculation wouldn't be accurate due to the strong correlation known to exist within the complex. It might be worth putting a material or something where DFT is known to succeed (and wavefunction or density matrix based methods can't perform)

Our response: These molecules are indeed not discussed and were there just to represent what can be done by FCI, Q-FCI or DFT, and Q-DFT in the future. We agree that DFT will not give accurate ground-state energy for FeMoco, however it could be used for geometry optimization to determine important structure (equilibrium geometry), as done commonly in the community even if DFT fails for strongly correlated systems. Nevertheless, the referee's comment convinced us that putting molecules in this figure is actually misleading. Also because the size of the systems that can be treated by FCI or DFT will depend on several factors, for instance the size and nature of the chosen basis set. Therefore, we decided to remove the molecules from the figure.

The referee writes:

Figure 4: The FCI calculation for H8 is cheap enough that it might be worth sampling a little bit more frequently such that the curve in the top panel is smooth (a few edges appear near the equilibrium distance and around 2.25\AA)

Our response: We will do more FCI calculations on H$_8$ to get smoother curves.

The referee writes:

  • On page 9 under Figure 3 there is the statement "For now on..." $\rightarrow$ maybe meant to be "From now on..."
  • Typo in the first line of Section 3.3 "The validity and applicability of the concept being proved" $\rightarrow$ proven instead of proved

Our response: We thank the referee for pointing these out, we will fix them in the revised manuscript.

---

## Round 2 · Referee Report · Anonymous (Referee 2) · 2022-9-22

Strengths

  1. Writing
  2. Figures
  3. Vision

Weaknesses

  1. Estimate of classical algorithms

Report

In "Toward Density Functional Theory on Quantum Computers?" (DFT), the authors propose a method to solve non-interacting Kohn-Sham equivalents of many-body physics problems with a quantum computer, supposing that the Kohn-Sham (KS) potential can be found from some other algorithm at least approximately. The algorithm leads to a $O(N^2)$ algorithm which is lower than the $O(N^3)$ that is the formal cost to solve a matrix and often quoted in KS-DFT. In contrast to several other works, this algorithm is focused on solving the Kohn-Sham system which is non-interacting and therefore cheaper. The paper speculates that the quantum computer could be used to determine the solution faster and a variational quantum eigensolver is used. A complete demonstration of the method is shown on example problems, albeit on chains of hydrogen atoms (single orbital).

The authors correctly state that diagonalizing a generic matrix on the classical computer is formally $O(N^3)$. However, the classical algorithms in LAPACK, with the exception of one step that is only computed once, are essentially $O(N^2)$ for the following reasons. The decomposition of a general matrix has the proceeding broad steps: 1) The matrix is converted to an upper (or lower) Hessenberg form. The matrix now contains all zeros above (or below) a specified sub-diagonal. [$O(N^3)$] 2) An implicitly restarted QR decomposition then determines the solution to numerical accuracy. I say a QR decomposition for simplicity since this is how these types of problems are generally taught, but this falls under the general class of a generic Schur decomposition, so let's extend the discussion to more than just eigenvalues, but these steps will apply equally there. [$O(N^2)$] The reason that the second step costs only $O(N^2)$ is due to the properties of the Hessenberg form (one can see this by supposing the zeros in the Hessenberg form do not need to be computed which is easy to write and can be stored appropriately). Effectively, all matrix multiplications beyond the initial set-up of the problem in step #1 are then $O(N^2)$ which is less than the standard matrix multiplication estimate. The bulk of the operations take place in step #2 and the first step only needs to be run once.

The first step is admittedly $O(N^3)$ to generate the Hessenberg form; however, I would allege that this first step is quite efficient and involves no recursive polynomials. At the end of the computation, some additional quantities may be required, but I am specifically thinking of those that require matrices multiplied onto vectors, only costing $O(N^2)$. Any of these steps also only needs to be performed once, as the Hessenberg matrix can be copied classically at essentially no cost.

The quantum algorithm's complexity of $O(N^2)$ is the same as the bulk of the classical algorithm, also $O(N^2)$. The manuscript goes into careful detail on how to perform what is essentially the second step on the quantum computer, and I found no obvious error there. The paper only briefly remarked that the wavefunction preparation to say that it was efficient, but the question is whether this step is better than the classical preparation. I note two points that are contrasted for the initialization: 1) The initialization step is run in each iteration of the algorithm here while the classical algorithm only runs this step once since there is a copy operation available classically 2) While the classical initialization step is $O(N^3)$, it is not clear how to compare that to the operations required on the quantum computer and it depends on what algorithm is used here (which is perhaps not referenced explicitly for generality in the text). As I said before, I would consider the classical step efficient and would need to see a direct comparison between this and the quantum preparation to say more. Effectively, the paper is comparing the initialization step in the classical solver with the major operations in the quantum algorithm. I think a reason for comparing these two steps is probably necessary since one could reason that since the wavefunction preparation on the quantum computer is efficient, it could also be said that the classical initialization is also inexpensive in practice.

To be complete, I suppose that I should also discuss how fast the classical algorithm will converge. The summary of the discussion so far is that, I think:

CLASSICAL: $N^3 + p*N^2$. (for $p$ iterations of the polynomial recursion for the decomposition)

QUANTUM: $M(pf(N)+1)*N^2$ (for measurements required to determine the variational parameters and $p$ updates of the classical parameters $\theta$ in the VQE and $f(N)$ includes time to find the classical parameters )

On its face, the quantum algorithm scales better (but I allege that the argument above and decades of efficiency with such algorithms about a single $O(N^3)$ step above seriously tilts towards the classical algorithm). The only question is how large the prefactors $p$ (one can perhaps give some charity that $p$ is equal in either the classical or quantum cases, although I would expect the quantum case to be somewhat larger) and $M$ should be. It is true that the classical algorithm converges as a polynomial of the original matrix. This often looks exponentially convergent, but I suspect it is super-linear (and I can not find a reference on that exactly right now, so let us just agree it is very fast if not exactly how fast). With degenerate eigenvalues, it might cost a few iterations, but without a concrete problem more discussion probably is not useful. To contrast that with the VQE used in this paper, we know that the determination of the classical parameters for the wavefunction is NP-hard ("Training variational quantum algorithms is NP-hard" from Phys. Rev. Lett. in 2021). I think this conclusively demonstrates that the quantum algorithm should be expected to be harder than the classical algorithm asymptotically assuming no additional structure of the original matrix. The prefactor on the quantum algorithm should be exponentially difficult while the classical algorithm is exponentially (or super-linear or something like it) suppressed. So, the scaling may not tell the whole story here and it would need to be stated explicitly where the crossover between the two algorithms would be to say more. I suspect the quantum algorithm might suffer from always being slower because of $M$, essentially, and the cost trade-off may not be worth the inclusion of errors in the values after that. The authors may have an argument with regards to the storage of extra precision on the resulting answer, but this would increase $M$ even more dramatically as opposed to simply paying the extra cost classically and writing a mostly small program. I also note that there are libraries capable of handling the extra numerical precision with some cost.

To put a conclusory statement on this, One might worry that $p$ is on the order of $N$ (the matrix size) and that the classical algorithm is indeed something like $N^3$. Unfortunately, this is not the case, especially for matrices with gaps between eigenvalues. Meanwhile, one would hope that $M$ would be small to keep the $N^2$ scaling, but as demonstrated in the paper, $M$ >> $N$ and should be expected to grow ($M$ >>> $N$ or more), with more than 1 million shots required for far fewer grid points. Again, $M$ should reflect the NP-hard aspects of finding those parameters. I do note that potentially one could use a shadow tomography to drastically reduce the number of measurements (for example, during the discussion on page 11, suppressing the exponential factors), but it is not immediately clear how to do that.

Having said all of that, I do find solutions of non-interacting problems on the quantum computer to be useful. There might be an opportunity to salvage this work with the core quantum algorithm. It is nice to know that the noisy quantum computer has a clean $O(N^2)$ implementation.

If the authors can situate their new algorithm in regards to the argument above or recast that step in a new way, then I would be happy to reconsider the main point presented here. I would further be happy if the non-interacting solver could either be developed for more general use-cases or if it could be shown how to extend it beyond the case of variational solvers. As always, this was what I gleaned from the paper. If the authors have a different view, then please let me know.

Requested changes

1) I think I agree that citations could appear in a few more places. To make this tangible, I will give a list of where I think the future readership would appreciate an extra citation: A) Eq. (3)...I believe a citation to Kohn and Sham's paper from 1964 would have this expression. B) James Whitfield has several papers that deal with DFT. It might be wise to cite him in some places

Computational complexity of time-dependent density functional theory, New J. Phys. Solver for the Electronic V-Representation Problem of Time-Dependent Density Functional Theory, JCTC Measurement on quantum devices with applications to time-dependent density functional theory, Frontiers in Physics

There are also a few other in the literature from other authors. I think in one of Preskill's recent papers with kernel-based Shadow tomography and machine learning, there was some application to DFT.

2) Around Eqs. (6-8), to emphasize the generality of the method, I would also cite the Bravyi-Kitaev transformation which accomplishes the same overall goal in a different context. 3) Eq. (9) could use the same energy symbol as Eq. (3), although I thought it was clear as written. 4) I want to know more about the statement: "This flavor of DFT appears optimal for quantum computers." when referring to SOFT. 5) The authors should clarify their invocation of the HHL algorithm. Ewin Tang has demonstrated that a classical algorithm is just as fast as HHL, rendering this algorithm mostly useless, to the dismay of the entirety of the quantum machine learning community. The authors should feel no remorse for this collapse since the leaders in the field misled everyone without proper correction. If the authors wish to still use the HHL algorithm, I might suggest expanding further on what is meant there. 6) As a light continuation of the major critique above, I note that the example chosen in the text to diagonalize a matrix is $O(N)$ since it is one-dimensional and one can take advantage of the tridiagonal structure of the geqrf function (or similar) in LAPACK. 7) I found the writing very good and I found no errors with the expressions. 8) With regards to the exponential size of the Hilbert space and the number of measurements towards the end, the authors may wish to read about Shadow Tomography from Scott Aaronson since that method answers some of the larger questions posed there affirmatively. 9) I noticed the space between the last word of the title and the interrogation point. The French package in Latex generates an extra space before the colon, but I am not sure if the same is happening here.

  • validity: low
  • significance: poor
  • originality: ok
  • clarity: high
  • formatting: excellent
  • grammar: perfect

Author:  Bruno Senjean  on 2022-10-19  [id 2932]

(in reply to Report 2 on 2022-09-22)

The referee writes:

In "Toward Density Functional Theory on Quantum Computers?" (DFT), the authors propose a method to solve non-interacting Kohn-Sham equivalents of many-body physics problems with a quantum computer, supposing that the Kohn-Sham (KS) potential can be found from some other algorithm at least approximately. The algorithm leads to a O(N^2) algorithm which is lower than the O(N^3) that is the formal cost to solve a matrix and often quoted in KS-DFT. In contrast to several other works, this algorithm is focused on solving the Kohn-Sham system which is non-interacting and therefore cheaper. The paper speculates that the quantum computer could be used to determine the solution faster and a variational quantum eigensolver is used. A complete demonstration of the method is shown on example problems, albeit on chains of hydrogen atoms (single orbital). The authors correctly state that diagonalizing a generic matrix on the classical computer is formally O(N^3). However, the classical algorithms in LAPACK, with the exception of one step that is only computed once, are essentially O(N^2) for the following reasons. The decomposition of a general matrix has the proceeding broad steps: 1) The matrix is converted to an upper (or lower) Hessenberg form. The matrix now contains all zeros above (or below) a specified sub-diagonal. [O(N^3)] 2) An implicitly restarted QR decomposition then determines the solution to numerical accuracy. I say a QR decomposition for simplicity since this is how these types of problems are generally taught, but this falls under the general class of a generic Schur decomposition, so let's extend the discussion to more than just eigenvalues, but these steps will apply equally there. [O(N^2)] The reason that the second step costs only O(N^2) is due to the properties of the Hessenberg form (one can see this by supposing the zeros in the Hessenberg form do not need to be computed which is easy to write and can be stored appropriately). Effectively, all matrix multiplications beyond the initial set-up of the problem in step #1 are then O(N^2) which is less than the standard matrix multiplication estimate. The bulk of the operations take place in step #2 and the first step only needs to be run once. The first step is admittedly O(N^3) to generate the Hessenberg form; however, I would allege that this first step is quite efficient and involves no recursive polynomials. At the end of the computation, some additional quantities may be required, but I am specifically thinking of those that require matrices multiplied onto vectors, only costing O(N^2). Any of these steps also only needs to be performed once, as the Hessenberg matrix can be copied classically at essentially no cost.

Our response We thank the referee for his relevant and detailed comment on matrix diagonalization on classical computers. We acknowledge that this is indeed true, though most of the cost remains on the tridiagonalization step of the symmetric Hamiltonian matrix that requires complexity 2N^3/3 and which generally proceeds with the Householder transformation. The following steps are of complexity O(N^2). However, what is of most importance to us is to overcome the upper bound that is still O(N^3). Indeed, it could appear unfair to compare the entire efficiency of a newly born quantum algorithm to classical eigensolver algorithms being, as mentioned by the referee, optimized over decades [see for instance Golub and van der Vorst, “Eigenvalue computation in the 20th century, Journal of Computational and Applied Mathematics, 123, 35-65 (2000)]. We do not claim that our algorithm is optimal, and we are absolutely convinced that several (existing or to-be-developed) improvements are possible. A more detailed discussion is provided below.

The referee writes

The quantum algorithm's complexity of O(N^2) is the same as the bulk of the classical algorithm, also O(N^2). The manuscript goes into careful detail on how to perform what is essentially the second step on the quantum computer, and I found no obvious error there. The paper only briefly remarked that the wavefunction preparation to say that it was efficient, but the question is whether this step is better than the classical preparation. I note two points that are contrasted for the initialization: 1) The initialization step is run in each iteration of the algorithm here while the classical algorithm only runs this step once since there is a copy operation available classically 2) While the classical initialization step is O(N^3), it is not clear how to compare that to the operations required on the quantum computer and it depends on what algorithm is used here (which is perhaps not referenced explicitly for generality in the text). As I said before, I would consider the classical step efficient and would need to see a direct comparison between this and the quantum preparation to say more. Effectively, the paper is comparing the initialization step in the classical solver with the major operations in the quantum algorithm. I think a reason for comparing these two steps is probably necessary since one could reason that since the wavefunction preparation on the quantum computer is efficient, it could also be said that the classical initialization is also inexpensive in practice. To be complete, I suppose that I should also discuss how fast the classical algorithm will converge. The summary of the discussion so far is that, I think: CLASSICAL: N^3+p∗N^2. (for p iterations of the polynomial recursion for the decomposition) QUANTUM: M∗(p∗f(N)+1)∗N^2 (for measurements required to determine the variational parameters and p updates of the classical parameters θ in the VQE and f(N) includes time to find the classical parameters) On its face, the quantum algorithm scales better (but I allege that the argument above and decades of efficiency with such algorithms about a single O(N^3) step above seriously tilts towards the classical algorithm). The only question is how large the prefactors p (one can perhaps give some charity that p is equal in either the classical or quantum cases, although I would expect the quantum case to be somewhat larger) and M should be. It is true that the classical algorithm converges as a polynomial of the original matrix. This often looks exponentially convergent, but I suspect it is super-linear (and I can not find a reference on that exactly right now, so let us just agree it is very fast if not exactly how fast). With degenerate eigenvalues, it might cost a few iterations, but without a concrete problem more discussion probably is not useful. To contrast that with the VQE used in this paper, we know that the determination of the classical parameters for the wavefunction is NP-hard ("Training variational quantum algorithms is NP-hard" from Phys. Rev. Lett. in 2021). I think this conclusively demonstrates that the quantum algorithm should be expected to be harder than the classical algorithm asymptotically assuming no additional structure of the original matrix. The prefactor on the quantum algorithm should be exponentially difficult while the classical algorithm is exponentially (or super-linear or something like it) suppressed. So, the scaling may not tell the whole story here and it would need to be stated explicitly where the crossover between the two algorithms would be to say more. I suspect the quantum algorithm might suffer from always being slower because of M, essentially, and the cost trade-off may not be worth the inclusion of errors in the values after that. The authors may have an argument with regards to the storage of extra precision on the resulting answer, but this would increase M even more dramatically as opposed to simply paying the extra cost classically and writing a mostly small program. I also note that there are libraries capable of handling the extra numerical precision with some cost. To put a conclusory statement on this, One might worry that p is on the order of N (the matrix size) and that the classical algorithm is indeed something like N^3. Unfortunately, this is not the case, especially for matrices with gaps between eigenvalues. Meanwhile, one would hope that M would be small to keep the N^2 scaling, but as demonstrated in the paper, M >> N and should be expected to grow (M >>> N or more), with more than 1 million shots required for far fewer grid points. Again, M should reflect the NP-hard aspects of finding those parameters. I do note that potentially one could use a shadow tomography to drastically reduce the number of measurements (for example, during the discussion on page 11, suppressing the exponential factors), but it is not immediately clear how to do that.

Our response We thank the referee for this interesting discussion. There are many aspects raised by the referee which are true. Some of these, like the number of measurements and the NP-hard optimization, are also true when applying VQE to the fully-interacting electronic structure problem. In the Q-DFT case, they could also prevent any quantum advantage when using ensemble-VQE as a solver. In the following, we give a – hopefully complete – answer to the comments made by the referee. It seems that the referee is attempting to directly compare the O(N^3) and O(N^2) steps of the classical algorithm to the different steps of the quantum algorithm. We believe that this is not how one should proceed, as quantum algorithms are completely different from classical ones and a direct comparison is not possible and sometimes misleading. Indeed, on a classical computer the limitation step is to obtain the Hessenberg form of the matrix, which scales as O(N^3). In Q-DFT, the preparation step is actually very easy and scales as a polynomial of log(N) [O(log(N)^(x+1)], where the log(N) corresponds to the number of gates of one layer of the hardware efficient ansatz, and the polynomial factor x corresponds to how the number of layers scales with the number of qubits. In our manuscript, we estimated that the number of layers scaled linearly [x = 1] with the number of qubits, but even if this is not true, increasing x would not matter much to the overall scaling. The bottleneck of the quantum algorithm is the number of repetitions (measurements, iterations, number of terms in the Hamiltonian) required to estimate the expectation values. On a classical computer, this is not an issue as the states are indeed stored. To summarize, classically the first step is a tridiagonalization process which scales as O(N^3) followed by steps scaling as O(N^2) for the diagonalization, while the quantum computer has to realize a huge number of O(log(N)^(x+1)) operations. This ‘huge’ number is the bottleneck, and we can estimate how it scales with the system size.

Taking the notations of the referee, the entire algorithm would scale as: M * P * f(N) * Nterms where M denotes the number of measurements per expectation value, P the number of iterations of the ensemble-VQE, f(N) the time execution of the circuit (gate complexity) and Nterms the number of terms of the Hamiltonian (number of expectation values to measure). • f(n): As discussed above, f(N) = O(log(N)^(x+1)), x being the power of the scaling of the number of layers with respect to the number of qubits. • Nterms: In the worst case scenario, the Hamiltonian has N^2 terms, and in the best case scenario O(N) terms only due to sparsity. • M: the number of measurements required to achieve an error e is directly proportional to the square of the 1-norm as M ≈ (sum_i |h_i|)^2 / є^2, where sum_i |h_i| is the 1-norm of the Hamiltonian H = sum_i h_i P_i and є is the error. It is generally assumed that the 1-norm scales in between O(N) and O(N^3) depending on the representation of the Hamiltonian. Several different approaches have been used to reduce the 1-norm, such as factorization techniques or the use of localized orbital basis, such that the scaling of the 1-norm is between O(N) and O(N^2). In the context of Q-DFT, the Hamiltonian is different as it represents only log(N) interacting pseudo-orbitals and has N^2 terms (or O(N) terms is sparse). Hence, we assume that the 1-norm is not a limiting factor here and that it should also scale as a polynomial of log(N). Especially, as we do not need chemical accuracy, the error of sampling can be larger and we will need less measurements. We therefore disagree with the referee when he/she says that the number of measurements might be exponential. However, the number of iterations might be, see below. • P: Indeed, optimizing VQE is NP-hard, even for a problem with a logarithmic number of qubits, and barren plateaus are also known to be present when using hardware efficient ansatz. This is certainly the most difficult part to deal with when using VQE-type algorithm. However, in practice we can reach some approximate states and orbital energies with a finite number of iterations. Finding efficient classical optimiser for VQE is a very active field of research. But still, VQE-based algorithms will certainly not be the most optimal quantum solvers in the future, and we are currently investigating the use of Quantum Phase Estimation or the Iterative Phase Estimation Algorithm rather than ensemble-VQE to solve Q-DFT.

To summarize, the overall scaling is as follows: Best-case scenario: M * P * f(N) * Nterms → P * O(log(N)^3 * N / є^2), with x = 1, 1-norm → O(log(N)) and Nterms → O(N) Worst-case scenario: M * P * f(N) * Nterms → P * O(log(N)^5 * N^3 / є^2) with x = 4, 1-norm → O(N) and Nterms → O(N^2)

Note that for the worst case scenario, we assumed x = 4 but more investigations are required and are left for future work. We also set a high bound of O(N) for the 1-norm. That being said, the measurements of all the number of terms are completely independent from each other. This has some importance as future quantum computers might as well be parallelized, such that the number of measurements wouldn't be a limiting step anymore. This strongly contrast with the limiting step of QR decomposition used in classical computer, as the Householder algorithm to obtain the tridiagonal form is iterative and as such impossible to parallelize. Hence, considering quantum parallelization, the only limiting factor is the circuit depth and the number of iterations of the classical solver, i.e. Best-case scenario: P * O(log(N)^2) Worst-case scenario: P * O(log(N)^5)

To conclude, without quantum parallelization, the worst-case scenario do not lead to any improvement over the classical algorithm. However, the best-case scenario seems to benefit from a smaller upper bound. Considering quantum parallelization, both cases scale as a polynomial of log(N), although optimizing VQE is in principle NP-hard. Finally, note that in contrast to Q-SOFT, one has to reconstruct the real-space density on the classical computer, which is a N^3 operation.

The referee writes

Having said all of that, I do find solutions of non-interacting problems on the quantum computer to be useful. There might be an opportunity to salvage this work with the core quantum algorithm. It is nice to know that the noisy quantum computer has a clean O(N^2) implementation. If the authors can situate their new algorithm in regards to the argument above or recast that step in a new way, then I would be happy to reconsider the main point presented here. I would further be happy if the non-interacting solver could either be developed for more general use-cases or if it could be shown how to extend it beyond the case of variational solvers. As always, this was what I gleaned from the paper. If the authors have a different view, then please let me know.

Our response We thank the referee for his positive remarks. We hope that we have clarified some aspects of the scaling of our algorithm, and we are happy to discuss it further if required. About the extension beyond variational solvers, this is currently under investigation and left for future work. For more general use-cases, our algorithm can actually be applied to any matrix diagonalization.

The referee writes

Requested changes 1) I think I agree that citations could appear in a few more places. To make this tangible, I will give a list of where I think the future readership would appreciate an extra citation: A) Eq. (3)...I believe a citation to Kohn and Sham's paper from 1964 would have this expression. B) James Whitfield has several papers that deal with DFT. It might be wise to cite him in some places Computational complexity of time-dependent density functional theory, New J. Phys. Solver for the Electronic V-Representation Problem of Time-Dependent Density Functional Theory, JCTC Measurement on quantum devices with applications to time-dependent density functional theory, Frontiers in Physics There are also a few other in the literature from other authors. I think in one of Preskill's recent papers with kernel-based Shadow tomography and machine learning, there was some application to DFT.

Our response We thank the referee for his/her suggestion. Indeed the papers of James Whitfield are relevant to this work and they are now cited in the introduction. Concerning the article of Preskill and coworker, we believe the referee refers to “Provably efficient machine learning for quantum many-body problems” (arxiv:2106.12627). However, apart from a mention of related works applying machine learning for DFT in the appendix, we didn’t find a direct link to our manuscript. If the referee actually refers to another paper that we didn’t find, we would be happy to add it to our manuscript.

The referee writes

2) Around Eqs. (6-8), to emphasize the generality of the method, I would also cite the Bravyi-Kitaev transformation which accomplishes the same overall goal in a different context.

Our response We added the references next to the Jordan-Wigner transformation. Indeed, any transformation of our auxiliary Hamiltonian to a qubit-Hamiltonian is valid.

The referee writes

3) Eq. (9) could use the same energy symbol as Eq. (3), although I thought it was clear as written.

Our response The energy in equations 3 and 9 (now 4 and 10 in the revised manuscript) are actually different. Indeed, in equation 3, the left-hand site energy is E_0, which is the ground-state energy of the many-body interacting problem. The latter is determined from the energy of the non-interacting KS problem, denoted by \mathcal{E}^{KS}, itself determined by adding together the energy of the occupied KS orbitals denoted by {\varepsilon_k}. Those {\varepsilon_k} are the ones obtained by solving equation 9.

The referee writes

4) I want to know more about the statement: "This flavor of DFT appears optimal for quantum computers." when referring to SOFT.

Our response The cost for solving one iteration of the self-consistent KS equations on the quantum computer is the same for both real-space DFT and SOFT. But for real-space DFT, the real-space density is required in the self-consistent loop, and it has to be calculated on the classical computer which scales as O(N^3), hence the statement above. However, we agree that this statement would be more understandable if it were mentioned later, i.e. after talking about the reconstruction of the real-space density, in the discussion of the scaling. We have therefore removed this sentence, and we have provided a more detailed discussion in the section “Discussion on the numerical efficiency” which we hope will make this statement clearer. Note that this section now also includes several aspects of our comparison between the classical and quantum algorithm scalings.

The referee writes

5) The authors should clarify their invocation of the HHL algorithm. Ewin Tang has demonstrated that a classical algorithm is just as fast as HHL, rendering this algorithm mostly useless, to the dismay of the entirety of the quantum machine learning community. The authors should feel no remorse for this collapse since the leaders in the field misled everyone without proper correction. If the authors wish to still use the HHL algorithm, I might suggest expanding further on what is meant there.

Our response We agree with the referee. We still would like to cite this article because the mapping to a logarithmic number of qubits is also done in their case, but of course we are not considering the same problem and we are not using the HHL algorithm at all. We reformulated the sentence as follows: “Such a mapping to a logarithmic number of qubits is analog to the mapping used in the Harrow-Hassidim-Lloyd algorithm, although the problems to solve are completely different.”

The referee writes

6) As a light continuation of the major critique above, I note that the example chosen in the text to diagonalize a matrix is O(N) since it is one-dimensional and one can take advantage of the tridiagonal structure of the geqrf function (or similar) in LAPACK.

Our response We understand the point of the referee and we agree. However, our example is a simple proof of concept. The systems that are usually treated by DFT are far from one-dimensional, and they are the one on which Q-DFT would be applied in the future. If the referee do not mind, we will not focus on the scaling of the classical algorithm for the system studied in the paper. However, a more detailed discussion on the general cases is added to the section “Discussion of the numerical efficiency”.

The referee writes

7) I found the writing very good and I found no errors with the expressions.

Our response We thank the referee for his/her very positive comment.

The referee writes

8) With regards to the exponential size of the Hilbert space and the number of measurements towards the end, the authors may wish to read about Shadow Tomography from Scott Aaronson since that method answers some of the larger questions posed there affirmatively.

Our response We thank the referee for his/her suggestion, as we didn’t think about it. Indeed, this strategy answers the problem of estimating every 4^M expectation values with only poly(M) measurements of our states. We have added a sentence and the reference to shadow tomography in the section “Discussion of the numerical efficiency” as well as in the conclusion.

The referee writes

9) I noticed the space between the last word of the title and the interrogation point. The French package in Latex generates an extra space before the colon, but I am not sure if the same is happening here.

Our response That would have been a good guess, unfortunately this was just a typical french typo… !

---

## Round 4 · Author Response

We would like to resubmit our manuscript entitled "Toward Density Functional Theory on Quantum Computers?".
We hope that the revised manuscript answers to all the questions, comments and demands raised by the two referees.
We hope that the revised manuscript answers to all the questions, comments and demands raised by the two referees.

---

## Round 4 · List of Changes

- New references have been added and discussed in the introduction
- "practical" has been added in front of "quantum advantage" in few places
- Bravyi-Kitaev reference has been added to the Jordan-Wigner one
- Molecules have been removed from Figure 1
- Figure 2 has been added, and shows the flowchart of Quantum-FCI and Quantum-DFT using VQE as a solver
- 10^4 shots state-vector simulations have been performed in addition to 10^5 and 10^6 shots, and the new results are discussed in the main text. Figures 4 and 5 have been modified accordingly.
- Extensive changes have been made to the section "Discussion on the numerical efficiency" to answer to Referee 2's comments. The section is now much more detailed, with new references (mentioning the 1-norm, shadow tomography, NP-hard optimization of VQE, ...)
- Fault-tolerant and non-variational solvers (such as Quantum Phase Estimation or the Iterative Phase Estimation algorithms) are mentioned as a follow-up of this work
- Any typos or other small changes raised by the referees have been corrected accordingly.

---

## Round 5 · Author Response

We would like to resubmit a corrected version of our manuscript entitled "Toward Density Functional Theory on Quantum Computers?".
We thank the referees for their meaningful comments that helped to improve the manuscript significantly.
Sincerely Yours,
Bruno Senjean, Saad Yalouz and Matthieu Saubanère.
We thank the referees for their meaningful comments that helped to improve the manuscript significantly.
Sincerely Yours,
Bruno Senjean, Saad Yalouz and Matthieu Saubanère.

---

## Round 5 · List of Changes

- “is orthonormal” has been added below Eq. 15, as well as the following paragraph in the Method section: “As our algorithm is designed for an orthonormal basis set, we used the Löwdin symmetric orthonormalization of the atomic orbitals to get a basis of orthonormal atomic orbitals. In principle, this step will be circumvented by using already orthonormal basis sets such as plane waves or Daubechies wavelets. They usually require much more basis functions but this is not a problem for Q-DFT as they are mapped on only log2(N) qubits”
- On page 5, "expected" changed to "originally hoped"
- On page 6, "Consequently, hardware efficient ansatz..." changed to "Consequently, a hardware efficient ansatz..."
- On page 12: "...number of interacting pseudo-orbital..." changed to "...number of interacting pseudo-orbitals..."

---

## Editorial Decision

published